# A mechanism of stratospheric $O_3$ intrusion to atmospheric environment: a case study of North China Plain

Yuehan Luo[1], Tianliang Zhao[1,*], Kai Meng[2], Jun Hu[3], Qingjian Yang[1], Yongqing Bai[4], Kai Yang[1], Weikang Fu[1], Chenghao Tan[5,6], Yifan Zhang[7], Yanzhe Zhang[8], Zhikuan Li[1]

[1]Collaborative Innovation Center on Forecast and Evaluation of Meteorological Disasters, Key Laboratory of Aerosol-Cloud-Precipitation of China Meteorological Administration, Nanjing University of Information Science and Technology, Nanjing, 210044, China

[2]Key Laboratory of Meteorology and Ecological Environment of Hebei Province, Hebei Provincial Institute of Meteorological Sciences, Shijiazhuang, 050021, China

[3]Fujian Academy of Environmental Sciences, Fuzhou, 350011, China

[4]Hubei Key Laboratory for Heavy Rain Monitoring and Warning Research, Institute of Heavy Rain, China Meteorological Administration, Wuhan, 430205, China

[5]State Key Laboratory of Organic Geochemistry, Guangzhou Institute of Geochemistry, Chinese Academy of Sciences, Guangzhou 510640, Guangdong, China

[6]University of Chinese Academy of Sciences, Beijing 100049, China

[7]Xuchang Meteorological Service, Xuchang, 450003, China

[8]Ningxia Air Traffic Management Sub-bureau of CAAC, Yinchuan, 750009, China

*Correspondence to*: Tianliang Zhao (tlzhao@nuist.edu.cn)

**Abstract.** Stratosphere-to-troposphere transport results in the stratospheric intrusion (SI) of $O_3$ into the free troposphere through the tropopause folding. However, the mechanism of SI influencing the atmospheric environment with the cross-layer transport of $O_3$ from the stratosphere, free troposphere to the atmospheric boundary layer has not been elucidated thoroughly. In this study, a SI event over the North China Plain (NCP, 33–40°N, 114–121°E) during May 19–20, 2019 was taken to investigate the mechanism of the cross-layer transport of stratospheric $O_3$ with the impact on the near-surface $O_3$ based on the multi-source reanalysis, observation data and air quality modeling. The results revealed a mechanism of stratospheric $O_3$ intrusion to the atmospheric environment induced by an extratropical cyclone system. The SI with downward transport of stratospheric $O_3$ to near-surface layer was driven by the extratropical cyclone system with vertical coupling of "upper westerly trough-middle the Northeast Cold Vortex (NECV)-lower extratropical cyclone" in the troposphere. The deep trough in the westerly jet aroused the tropopause folding, and the lower stratospheric $O_3$ penetrated the folded tropopause into the upper and middle troposphere; the westerly trough was cut off to form a typical cold vortex in the upper and middle troposphere. The compensating downdrafts of the NECV pushed the further downward transport of stratospheric $O_3$ in the free troposphere; The NECV activated an extratropical cyclone in the lower troposphere, and the vertical cyclonic circulation governed the stratospheric $O_3$ from the free troposphere across the boundary layer top invading the near-surface atmosphere. In this SI event, the averaged contribution of stratospheric $O_3$ to near-surface $O_3$ was accounted for 26.77%. The proposed meteorological mechanism of vertical transport of stratospheric $O_3$ into the near-surface atmosphere driven by an

extratropical cyclone system could improve the understanding of the influence of stratospheric $O_3$ on atmospheric environment with implications for the coordinated control of atmospheric pollution.

Keywords: Stratospheric intrusion; Tropopause folding; Near-surface ozone; Extratropical cyclone system

## 1 Introduction

The prominence of tropospheric $O_3$ pollution is growing increasingly conspicuous in China (Chen et al., 2021; Li et al., 2020;
Luo et al., 2022). The widely concerned photochemical reactions and regional transport are the principal sources of tropospheric $O_3$ in air pollution. Many studies have ascertained the mechanism of these factors on tropospheric $O_3$ and proposed targeted control strategies for $O_3$ pollution (Ding et al., 2019; Li et al., 2019; Li et al., 2020; Liu et al., 2021; Verstraeten et al., 2015). In addition, approximately 90% of the atmospheric column $O_3$ abundance concentrates in the stratosphere. The stratosphere intrusion (SI) with the downward cross-layer transport of stratospheric $O_3$ is also considered
an essential natural source of tropospheric $O_3$ (Bourqui and Trepanier, 2010; Chen, et al., 2023a; Holton et al., 1995; Wild, 2007), which has been non-comprehensively understand with the effect mechanism and degree on atmospheric environment change.

Under specific configuration of atmospheric circulations, stratospheric $O_3$ could invade downwards through the tropopause along with the dry and cold stratosphere air with high potential vorticity (PV), increasing the tropospheric $O_3$ concentrations
with a contribution of about 300–700 Tg $O_3 \cdot yr^{-1}$ (Stevenson et al., 2006; Wild, 2007). As the primary pathway of stratosphere-troposphere transport (STT), SI is achieved by multiple synoptic-scale and mesoscale processes (Stohl et al., 2003). Ordinarily, SI is closely associated with the subtropical westerly jet, cut-off low pressure, tropopause folding, trough-ridge system, tropical cyclones, and convection systems, etc (Chen et al., 2022; Chen et al., 2023b; Ding and Wang, 2006; Holton et al., 1995; Jiang et al., 2015; Zhao et al., 2021). The deeply folded tropopause in the subtropical westerly jet
with a cold front passing the western United States induced near-surface $O_3$ increases (Langford et al., 2012). The widespread subsidence movement in the south of the subtropical jet in March 2010 brought about a significant increase of $O_3$ in the low troposphere in Hong Kong (Zhao et al., 2021). In June 2014, the peripheral downdrafts in deep convection of the large-scale tropical cyclone Hagibis directly drove the STT of $O_3$, prompting the stratospheric $O_3$ transport downwards and causing rapid abnormal enhancements in surface $O_3$ (Jiang et al., 2015).

SI exhibits greater prominence in the mid-latitudes with the prevailing subtropical westerly jet. In the mid-latitudes of the northern hemisphere, approximately 20–30% of the $O_3$ reserve in the troposphere is sourced from the stratosphere. (Lelieveld and Dentener, 2000). SI events were observed frequently in the western United States, the eastern Mediterranean, the Middle East, and East Asia (Akritidis et al., 2016; Langford et al., 2009; Lefohn et al., 2011; Wang, X. et al., 2020). In parts of the US, the contribution of SI to surface $O_3$ is almost comparable to that of local $O_3$ precursor emissions (Lin et al.,

2012; Pfister et al., 2008). In China, the tropospheric $O_3$ exhibited an annual increase of 0.78% $yr^{-1}$ on account of the SI processes (Verstraeten et al., 2015). The SI caused by a landing typhoon sharply increased the surface $O_3$ by 32% in the North China Plain (NCP) (Meng et al., 2022a). Evidently, the artificial uncontrollability of SI impact on surface $O_3$ complicates the prevention and control of ambient $O_3$ pollution. Therefore, clarifying the influence mechanism of SI on surface $O_3$ could improve the knowledge system on the formation causes of ambient $O_3$ pollution, which is of great concern for formulating effective prevention of $O_3$ pollution.

It is generally recognized that the SI processes are closely associated with subtropical westerly jet activities (Archer and Caldeira, 2008; Langford et al., 2012). The abnormal disturbances of the jet streams, which are the principal reason for the tropopause folding, could further develop to form a cold vortex over the mid-upper troposphere in mid-high latitudes (Martin, 2021; Satyamurty and Seluchi, 2007). The cold vortex regulates the weather and climate in the mid-high latitudes (Xue et al., 2022; Yang et al., 2017), and triggers vigorous convection, resulting in a variety of disastrous weather and anomalous climate (Tao et al., 2023; Xue et al., 2021). The Northeast Cold Vortex (NECV) is a typical cold low vortex system in East Asia (Lian et al., 2016; Xie and Bueh, 2017). The pronounced 500-hPa trough-ridge system of the subtropical westerly jet from the Caspian Sea to eastern China is considered the precursor of the NECV (Huang et al., 1997). And on account of the existence of the upper-level jet with the NECV, a significant horizontal wind vertical shear is aroused between the mid-lower stratosphere and the NECV, potentially inducing the stratosphere-troposphere exchange of air mass. The overall impact of the NECV-resulting mass exchange at the tropopause is the net import of $O_3$ from the stratosphere to the troposphere (Chen et al., 2014a; Chen et al., 2014b). However, numerous existing studies on the cold vortex system focus on its causal factors of formation and structure as well as its effects on precipitation and extreme weather (Fu et al., 2014; Gao and Gao, 2018; Lian et al., 2016; Tao et al., 2023; Xue et al., 2021), the mechanism of the cold vortex in mid-high latitudes driving the vertical cross-layer transport of stratospheric $O_3$ and the atmospheric environmental impact still lacks an unambiguous insight in need of thorough investigation.

Therefore, this study targeted the SI event driven by a typical NECV observed over the NCP during May 19–20, 2019 to investigate the mechanism of stratospheric $O_3$ intrusion affecting the atmospheric environment. Multi-source reanalysis data and the Weather Research and Forecasting/Chemistry (WRF-Chem) with integrated process analysis (IPR) are jointly applied to delve into the mechanism of the cold vortex system triggering SI and the SI effect degree on surface $O_3$. This study aims to elucidate the mechanism of the vertical structure configuration of cold vortex systems in mid-high latitudes driving $O_3$ cross-layer transport and quantify the influence on surface $O_3$, refining the systematic understanding of the stratosphere-free troposphere-atmospheric boundary layer (ABL) transport of air pollutants with the application implication for improving the response capabilities to in atmospheric environment change.

**2 Data and Methods**

## 2.1 Observational and reanalysis data

3-D meteorological data in this study were obtained from the 0.25° ERA5 reanalysis product (https://apps.ecmwf.int/datasets/) released by the European Centre for Medium-Range Weather Forecasts (ECMWF). According to previous studies, meteorological variables such as air temperature, wind, and humidity of ERA5 reanalysis data show good performance in both stratosphere and troposphere (Hersbach et al., 2020). The ground meteorological data were collected from 639 meteorological stations in the NCP and surrounding region from the China Meteorological Observation Network. The data of 3-D $O_3$ concentrations were provided by the Modern-Era Retrospective Analysis for Research and Applications version 2 (MERRA2, https://gmao.gsfc.nasa.gov/reanalysis/MERRA-2/data_access/) with a horizontal resolution of 0.5° × 0.625° and a temporal resolution of 3 hours, to characterize the vertical transport of the stratospheric $O_3$. The MERRA-2 data set assimilates TCO satellite retrievals from ozone monitoring instruments and stratospheric $O_3$ vertical profiles from microwave limb sounders after 2004 (Gelaro et al., 2017; Levelt et al., 2006; Waters et al., 2006). The MERRA2 $O_3$ enables the study of SI events because it is consistent well with ozonesondes and could realistically represent the temporal and spatial variations of $O_3$, especially in the lower stratosphere and near the tropopause (Wargan et al., 2015; Wargan et al., 2017). Hourly observed surface $O_3$ and CO concentrations retrieved from 440 observation sites of the National Air Quality Monitoring Network (http://www.mee.gov.cn/), with quality control based on China's National Standard of Air Quality Observation. To evaluate the reproducibility of the WRF-Chem simulations for the stratospheric $O_3$ intrusion, the portion ($O_3$S) of tropospheric $O_3$ concentrations originating from the stratosphere was applied to compare with our simulation results. The stratospheric tracer tagging method in global chemistry models was used to track the transport of stratospheric $O_3$ to the troposphere by releasing stratospheric tracers (Barth et al., 2012; Chang et al., 2023). The tracer was set to 1 above the tropopause, and only physically transported and chemically decayed in the troposphere without chemical productions (Chang et al., 2023; Ni et al., 2019). The $O_3$S in the troposphere was calculated by multiplying the concentrations of $O_3$ at the tropopause and the stratospheric tracers. The $O_3$S used in this study was obtained from the ECMWF Atmospheric Composition Reanalysis 4 (EAC4, https://ads.atmosphere.copernicus.eu/cdsapp#!/dataset) reanalysis dataset with a spatial resolution of 0.75°, temporal resolution of 3 h, and vertical divided into 60 model layers from 1000 hPa to 1 hPa. The EAC4 dataset produces optimized atmospheric chemical reanalysis results by assimilating multiple observations of atmospheric components, and is widely used in the field of atmospheric environment (Inness, et al., 2019).

## 2.2 Modeling setup and configuration

The meteorology-chemical online coupled model WRF-Chem with IPR was applied in this study. WRF-Chem has been demonstrated to perform well in the simulation and analysis of SI cases (Chang et al., 2023; Ni et al., 2019; Zhao et al., 2021). The 5-day simulations covering three nested domains started on May 16, 2019, with horizontal resolutions of 81 km, 27 km, and 9 km, respectively. The initial and lateral boundary meteorological conditions at 6-hour intervals are derived

from ERA5 reanalysis data, and the Multi-resolution Emission Inventory of China (MEIC, http://meicmodel.org/) provided the anthropogenic emission data. The domain settings are presented in Fig. S1, and the physical and chemical parameterization schemes used in the simulations are listed in Table S1 in the Supplementary Material.

Since the stratospheric chemistry is not included in the WRF-Chem, an upper boundary condition (UBC) scheme derived from the Whole Atmosphere Community Climate Model was used to provide the initial and boundary chemical conditions for the stratosphere in the model (Barth et al., 2012). The UBC scheme could generate all key chemical species in the stratosphere, enabling the WRF-Chem to simulate the stratospheric intrusion processes more accurately (Barth et al., 2012; Lamarque et al., 2012; Zhao et al., 2021). To diagnose the impact of stratospheric $O_3$ on surface $O_3$ during the SI caused by

the cold vortex system, two simulation experiments were devised. The control experiment ($CASE_{STRO3}$), with the upper boundary conditions of stratospheric $O_3$ in WRF-Chem (Barth et al., 2012), was used to simulate the real meteorological and atmospheric chemical fields during the SI. The sensitivity experiment ($CASE_{noSTRO3}$) used identical physical and chemical configurations as in $CASE_{STRO3}$. The stratospheric $O_3$ of the UBC scheme in $CASE_{noSTRO3}$ was turned off by setting $O_3$ concentrations to zero above the tropopause. The differences in $O_3$ simulation between the experiments $CASE_{STRO3}$ and

$CASE_{noSTRO3}$ are used to assess the quantitative effect of the stratospheric $O_3$ sources on surface $O_3$ in the SI event.

**2.3 Modeling validation**

The observed meteorological and chemical parameters, including air temperature at 2 m ($T_2$), wind speed at 10 m ($WS_{10}$), relative humidity at 2 m ($RH_2$), and $O_3$ concentrations, were used to assess the model performance in $CASE_{STRO3}$ (Fig. S2). We calculated the averages of meteorological elements and $O_3$ concentrations observed at all stations in the innermost

domain and the averages of simulated meteorological elements and $O_3$ concentrations in the model grids corresponding to the station locations to conduct the modeling validation. It can be found that the simulated $T_2$, $RH_2$, and $WS_{10}$ generally agree well with the observations, the index of agreements (IOA) are 0.92, 0.99, and 0.90, respectively, and the correlation coefficients (R) are higher than 0.92, which all meet the usability criteria (Wei et al., 2019; Zhang et al., 2018). Hourly observed and simulated $O_3$ concentrations are shown in Fig. S2 (d). The R, MB (mean bias), RMSE (root mean square

errors), and IOA of $O_3$ are 0.88, -8.25 %, 14.37, and 0.90, respectively, indicating that our simulation reasonably reproduced the variations of $O_3$ in $CASE_{STRO3}$. In Fig. S3, the differences of simulated surface $O_3$ between $CASE_{STRO3}$ and $CASE_{noSTRO3}$, which indicate the contribution of stratospheric $O_3$ to the near-surface atmosphere, were compared with the $O_3S$ of EAC4 reanalysis data and concluded that the simulations roughly captured the temporal and spatial variations of stratospheric $O_3$ reaching the near-surface layer during the SI. All these evaluations indicate that our simulations performed well in

reproducing the variations of $O_3$ and meteorological parameters during the SI process.

**3 Results and discussion**

## 3.1 Vertical configuration triggers O₃ cross-layer transport

The tropopause identifies the top of the troposphere and marks a separation with the stratosphere in terms of both thermal structure and chemical composition. SI events are characterized by the downward transport of stratospheric O₃-rich air, coinciding with high PV and low humidity cross the tropopause into the troposphere (Ganguly, 2012). The height with PV equal to 2 is widely used to indicate the position of the dynamical tropopause at mid-latitudes (Holton et al., 1995). The tropopause over the NCP presented the obvious variation over May 18–20, 2019 (Fig. S4), especially the tropopause folding with the contour at PV = 2 curving extremely downwards to below 400 hPa over the NCP region on May 19, 2019 (Fig. S4b). Figure 1 shows the latitudinal- and meridional-vertical structures of the dynamical tropopause on May 19, 2019 over the NCP. The tropopause noticeably folded over the area of 110–120E° and 39–42N°, dropping down to 500–600 hPa. In the tropopause folding region, strong downdrafts penetrated from the lower stratosphere to the near-surface layer, and the strong vertical velocity reaching up to 0.2 m·s⁻¹ was inclined to drive the downward transport of the stratospheric O₃ to the troposphere (Fig. 1). Correspondingly, the vertical O₃ distribution presented a tongue pattern with the O₃ concentrations exceeding 200 ppb poured into the troposphere along with the dry and cold air from the stratosphere (Fig. S4b), which indicating the SI influencing O₃ in the ambient atmosphere.

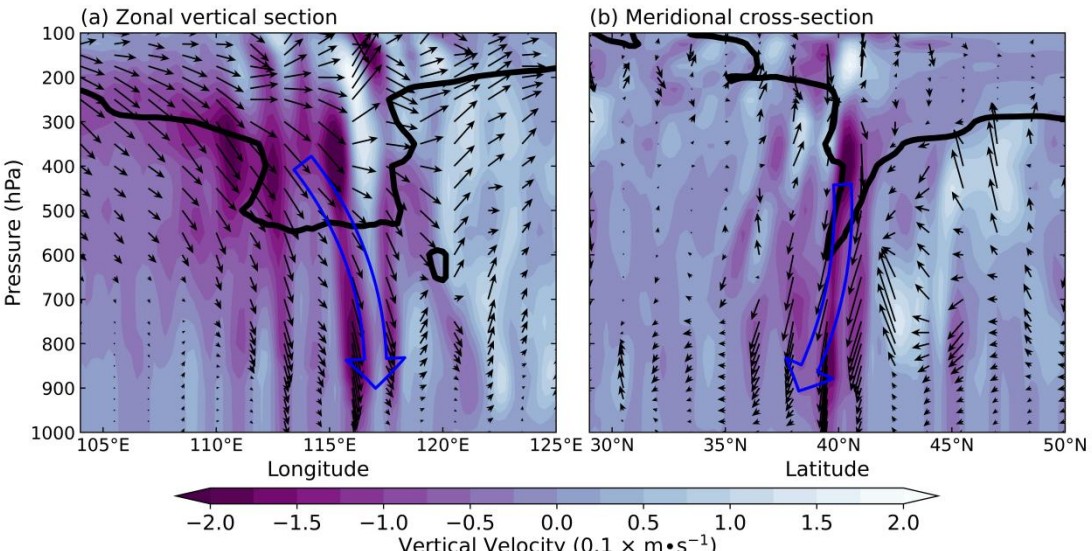

**Figure 1: The vertical sections of the vertical velocity (color contours; 0.1×m·s⁻¹) and wind vectors of the components of U and V with W × 50 from the ERA5 data averaged (a) over 32 °N–40 °N and (b) over 114 °E–121 °E in the NCP region on 19 May 2019. Black solid lines indicate the dynamical tropopause marked with the isolines at PV=2. The blue hollow arrows indicate the downdraft flows of SI in the troposphere.**

Figures 2 (a–c) and S5 presented the upper, middle and lower tropospheric atmospheric circulations at 200 hPa, 500 hPa, and 850 hPa with the temporal variations during the NECV period. At 16:00 LST (Local Standard Time in 24-hour system) on May 18, 2019, the shallow troughs were located over 95–100E° in the subtropical westerly jets at 200 hPa and 500 hPa (Fig. S5). The subtropical westerly jet evolved from zonal to meridional patterns with the trough moving eastward and deepening

southward in the westerly wave fluctuations, and the prominently deepening troughs moved to the northern NCP on May 19 (Fig. 2a–b). The deepening troughs at the upper and middle troposphere at 200 hPa and 500 hPa were cut off forming the cyclonic cold vortex system—the NECV during May 19–20, 2019 (Figs. 2 and S5). The NCP was situated at the periphery of the NECV with the compensating downdrafts, which could drive the vertical transport down in the free troposphere (Fig. 2). Moreover, the NECV in the upper and middle troposphere induced the extratropical cyclone in the lower troposphere at 850 hPa on May 19, 2019, and the strong northwest horizontal wind prevailed behind the extratropical cyclone in the NCP (Fig. 2). On May 20, the NECV moved northeastwards and lost control of the NCP (Fig. S5). Figure 2d presents the section of the vertical velocity from the NCP to the center of the extratropical cyclone (red solid line in Fig. 2c) on May 19. The intense rising motions presented near the center of the NECV and the extratropical cyclone, ranging from the near-surface layer (1000 hPa) up to 100 hPa. The sinking branch of the vertical circulation of the cold vortex and the lower-tropospheric cyclone from the upper troposphere (200–400 hPa) to the near- surface layer was situated over the NCP (Fig. 2d), governing the downward transport of the stratospheric $O_3$ to the near-surface layer. Namely, the atmospheric circulation presented the vertical configuration in troposphere with upper westerly trough, middle NECV and lower extratropical cyclone. The downdraft branch of the system opened up a vertical transport channel from the lower stratosphere to the surface, allowing stratospheric $O_3$ to invade the surface.

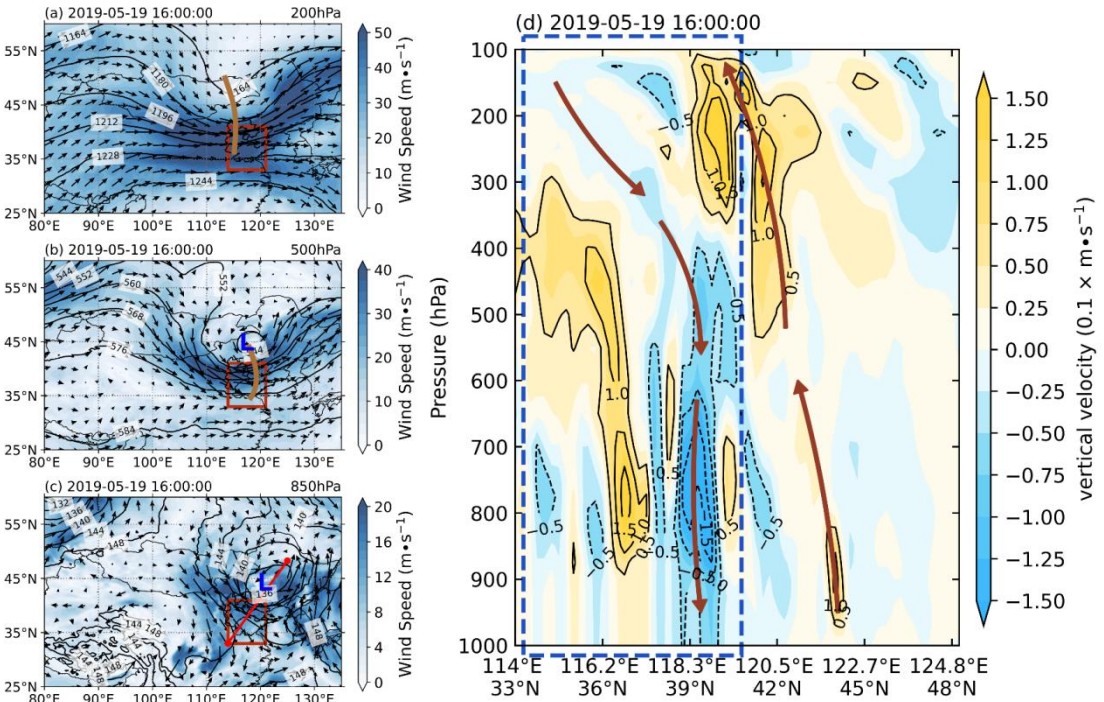

**Figure 2: Atmospheric circulation patterns of horizontal wind vectors from the ERA5 data at (a) 200 hPa, (b) 500 hPa, and (c) 850 hPa with brown lines indicating the troughs in the westerlies, and (d) vertical cross sections of vertical velocity (color contours, 0.1×m·s⁻¹) along the red straight lines in (c) at 16:00 LST on May 19, 2019. The red solid boxes in (a–c) and the blue dotted box in (d) mark the NCP region. The solid red line in (c) is perpendicular to the northwest wind direction over the NCP region passing through the extratropical cyclone center at 850 hPa, and the brown lines with arrows in (d) indicating vertical transport directions between the lower stratosphere to the lower troposphere.**

In the middle and lower troposphere, the downdrafts appeared over the northwest of the NCP and gradually reinforced with the development of the cold vortex and the induced extratropical cyclone (Fig. 3). The mountain airflow on the lee slopes of the plateau and mountains in western NCP could also contribute to strengthening the subsidence motion in the lower troposphere (Ning et al., 2018). Strong downdrafts in the lower troposphere, combining with the high turbulent diffusion in the convective boundary layer during the daytime, could forced the tongue plume with $O_3 > 150$ ppb to cross the boundary layer top extending to the near-surface layer from 9:30 to 15:30 LST on May 19 (Figs. 3c and d). Horizontally, the strong northwest wind at the rear of the NECV and near-surface cyclone induced the $O_3$-rich tongue plume to bend from the northwest to the southeast of the NCP (Figs. 2c and 3). Therefore, the stratospheric $O_3$ reached the surface of northwest NCP in the morning, gradually drifted along the direction of the airflow, and then reached the southeast NCP at night on May 19, 2019 (Figs. 3c–e). In this vertical configuration of atmospheric circulations of "upper westerly trough-middle NECV-lower extratropical cyclone" from the lower stratosphere to the near-surface layer (Fig. 2), the deep westerly trough with strong vertical wind shear triggered the tropopause folded downward, which was expected to induce the $O_3$-rich air mass in the stratosphere to pour into the middle and upper troposphere (Figs. 2–3). Subsequently, the compensating downdrafts at the periphery of the cold vortex circulation effectively transport the high concentrated $O_3$ from the upper troposphere downwards to the lower troposphere. The descending branch of extratropical cyclone in the lower troposphere further promoted the transport of stratospheric $O_3$ from the free troposphere to the ABL in the lower troposphere.

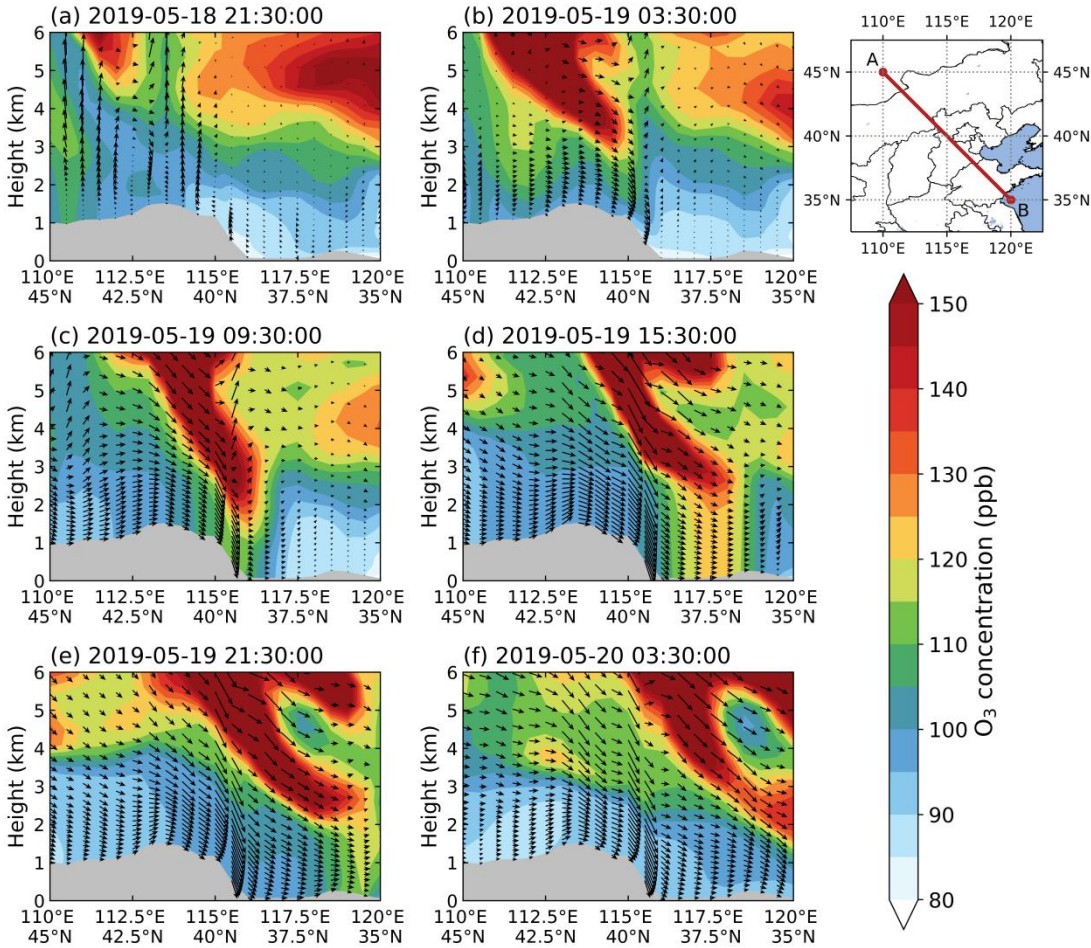

**Figure 3: Vertical sections of O₃ concentrations (ppb) and composite wind vectors from the MERRA2 data along the line AB in the middle and lower troposphere over the NCP with the topography marked with gray areas. The line AB crosses the entire NCP along the northwest wind direction.**

### 3.2 Impact of the SI on near-surface O₃

To investigate the impact of stratospheric O₃ on near-surface O₃ caused by the SI, the observed near-surface O₃ concentrations, the wind fields, and the 24-hour changes of air temperature at 950 hPa are shown in Fig. 4. Before the NECV formed on May 18, strong cold air was centered in the northeast-southwest zone in north China connecting with a slight cooling in the northwestern NCP, which was reflected by the 24-hour changes of air temperature at 950 hPa (Fig. 4a). Meanwhile, a strong warm easterly airflow prevailed over the eastern and southern NCP with lower O₃ concentrations, and the high O₃ concentrations existed over the convergence zone between weak and strong easterly winds in the northwestern NCP region. With the NECV formation on May 19, the extratropical cyclone in the lower troposphere was activated in the northern NCP and the Northeast China, and the northwest airflows at the rear of cyclone pushed the cold air mass southeastwards and prompted high O₃ in the northwestern NCP to spread downstream (Fig. 4b). On May 20, the NCP region was beyond of NECV control with decreasing O₃ concentrations (Fig. 4c).

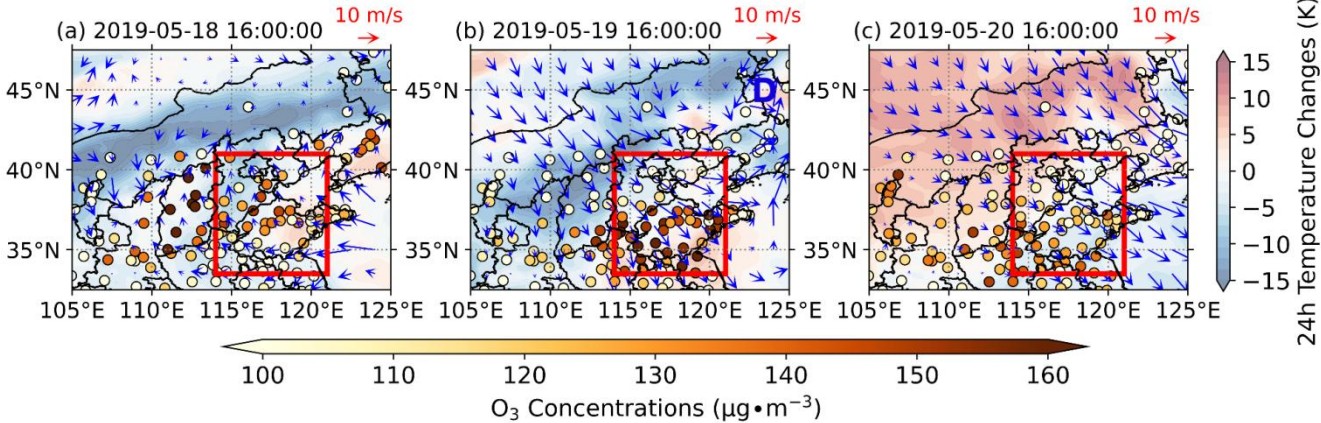

**Figure 4: Horizontal wind vectors and 24-hour changes of air temperature (shaded colors) at 950 hPa from the ERA5 data and the observed near-surface O$_3$ concentrations (color dots) at 16:00 LST on May (a) 18, (b) 19, and (c) 20, 2019. The red rectangles cover roughly the NCP region.**

Figure 5 shows the temporal variations of near-surface O$_3$ and CO concentrations and meteorological variables over the NCP. Due to the impact of cold air on the NCP, the rising air temperature required for the stronger photochemical reaction was not observed on May 19, 2019 (Fig. 5a). Precipitation in the NCP region could scavenge the O$_3$ precursors (Fig. 5a), which could modify the photochemical O$_3$ production (Sato et al., 2006; Yoo et al., 2014). Also, the dense cloud cover could reduce the solar radiation suppressing the O$_3$ production from photochemical reactions (Fig. 5b). Since the stratosphere is rich in O$_3$ and poor in CO, the homology between CO and O$_3$ is commonly regarded as an indicator to identify the source of O$_3$ (Moody et al., 1995; Parrish et al., 1998; Voulgarakis et al., 2011). The negative correlation between CO and O$_3$ implies that O$_3$ derived from the vertical downward STT. As can be found from Fig. 5c, with the formation of tropopause folding and the cold vortex on May 19 (Fig. S2), the regional CO concentrations in the NCP decreased rapidly, reaching nearly half of the initial level. Meanwhile, O$_3$ concentrations slightly increased. Remarkably, the opposite variations between CO and O$_3$ combined with the rapidly decreased T$_2$ and relative humidity (RH$_2$) could reflect the stratospheric O$_3$-rich air intrusion into the ground redistributing surface O$_3$ in the NCP region (Figs. 5a and c). In addition, triggered by the cold vortex in the upper and middle troposphere (Fig. 2), the extratropical cyclone dramatically increased the near-surface wind speed (WS$_{10}$) over the NCP (Fig. 5b), with the maximum regional averaged wind speed exceeding 7 m·s$^{-1}$. The strong northwest winds of the cyclone could enhanced the diffusion of O$_3$, which were beneficial to the decrease of local near-surface O$_3$ concentrations in the SI event. Although stratospheric O$_3$ intensely invaded the near-surface over the NCP region, the near-surface O$_3$ was strongly diffused downstream by the northwest wind. Meanwhile, the horizontal diffusion also prevented O$_3$ from the stratosphere from accumulating over the NCP (Figs. 2a–c and Fig. 4). Also, the routine ground observations cannot distinguish whether O$_3$ comes from the stratosphere or local generation. However, the near-surface O$_3$ observed on May 19 was slightly higher than the previous day under such facilitated diffusion conditions (Fig. 5c), which proves that the SI exerted additional contributions on the near-surface O$_3$ over the NCP region. Additionally, the changes in observed meteorological and environmental elements from the representative sites SJZ and JN in the NCP (The red dots in Fig. S3) were examined in Fig.

S6. The results showed the distinct characteristics compared with the regional averages that the diurnal cycles of $O_3$ concentrations were disturbed with the $O_3$ peaks by the SI event . The SJZ in the northwest NCP received stratospheric $O_3$ earlier and reached the spike at 10:00 LST on May 19. Then the $O_3$ concentrations gradually decreased under the influence of strong winds but still maintained a high level in the early morning of May 20. The JN in the southeast NCP was affected by the stratospheric intrusion later. While under meteorological conditions conducive to the dissipation of pollutants (wind speed up to 8 m·s$^{-1}$), higher $O_3$ concentrations than the previous day were still observed, reflecting the additional contribution of stratosphere intrusion to near-surface $O_3$.

Overall, the vertical configuration of 1) The abnormal disturbance of the subtropical westerly jet induced a deep westerly trough between lower stratosphere and upper troposphere, and the strong downdrafts on the south side of the trough triggered the tropopause folding, allowing stratospheric $O_3$ to invade into the upper and middle troposphere (Blackmon et al., 1977). 2) The deepening troughs were cut off forming the cold vortex in the upper and middle troposphere, the sinking branch of the cold vortex vertical circulation driven stratospheric $O_3$ to continue to be transported downward to the lower troposphere. 3) The cold vortex excited an extratropical cyclone in the lower troposphere, and the compensating downdrafts on the periphery of the cyclone could pushed $O_3$ from the stratosphere to invade the near-surface, which could be a mechanism of stratospheric $O_3$ intrusion affect the atmospheric environment in the following study.

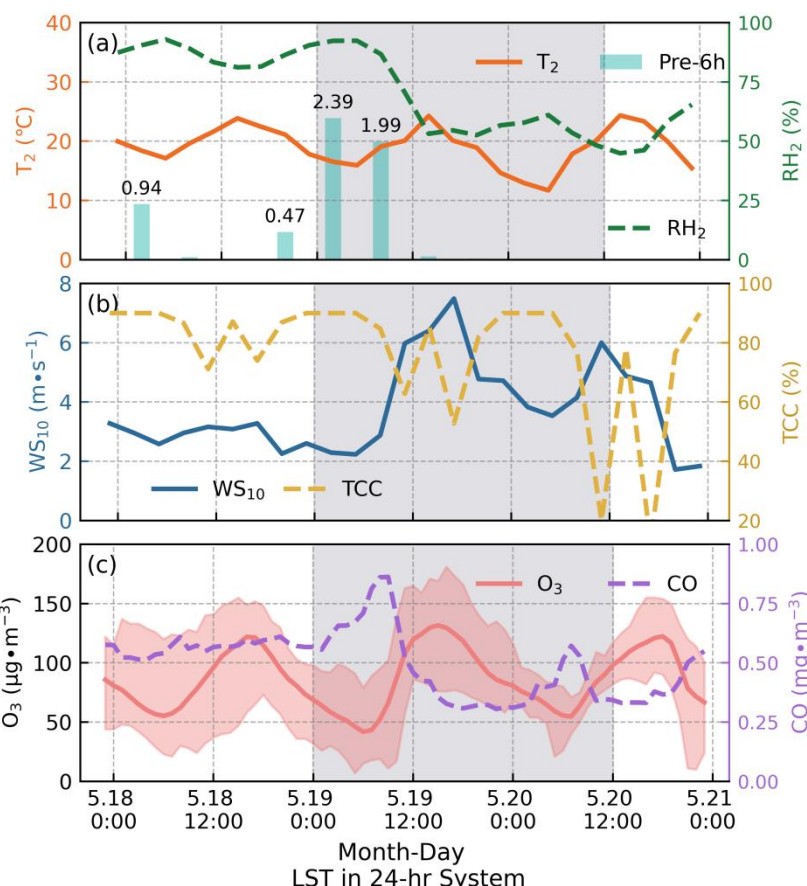

**Figure 5: Temporal variations of regionally averaged (a) $T_2$, accumulated precipitation of 6 hours (Pre-6h, units: mm), RH$_2$, (b) WS$_{10}$, and total cloud cover (TCC), (c) near-surface $O_3$ and CO concentrations from the observations in the NCP region. The**

**shaded areas mark the periods of the SI to the near-surface layer.**

Quantification of the influence of NECV-driven SI on near-surface $O_3$ is crucial to deepen the understanding of SI process in $O_3$ pollution. To quantify the contribution of stratospheric $O_3$ to the near-surface layer during the SI event, we adopted the WRF-Chem model with IPR. The IPR can be used to quantitatively analyze the contributions of physical and chemical processes to $O_3$. Five principal processes are considered in WRF-Chem, including subgrid convection (CONV), chemical conversion (CHEM), vertical mixing (VMIX), horizontal advection (ADVH), and vertical advection (ADVZ) (Yang et al., 2022). The ADVH and the ADVZ are caused by the delivery of horizontal and vertical winds, and the ADVZ can be used to characterize the intrusion of the stratospheric $O_3$ into the ABL.

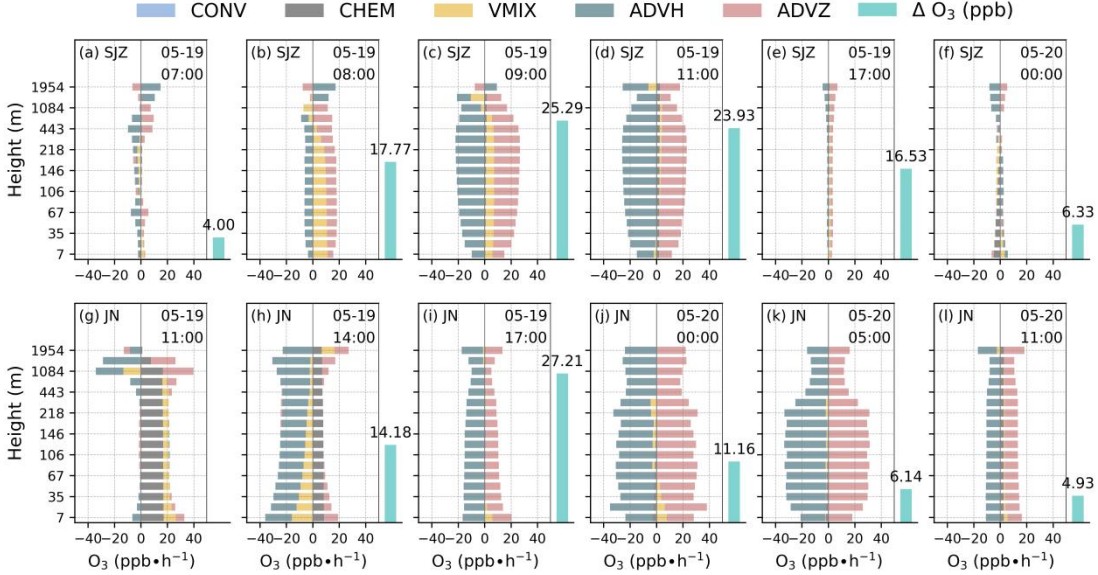

**Figure 6: Temporal variations of contributions of atmospheric physical and chemical processes to $O_3$ from the simulation CASE$_{STRO3}$ in the ABL in representative cities SJZ and JN in the NCP during the SI event.**

In this study, we selected two sites of SJZ and JN (the red dots in Fig. S3) respectively in the northern and southern NCP as representatives to analyze the results of IPR (Fig. 6). It is noteworthy that the ADVH and ADVZ presented the generally reverse contributions in complementary ways, and the positive ADVZ contributions dominated the enhancements of near-surface $O_3$, indicating the vertical $O_3$ transport of the SI associated with the extratropical cyclone induced by the NECV (Figs. 3–6). The compensating subsidence branch of the extratropical cyclone further drove the stratospheric $O_3$ across the top of the ABL to the near-surface layer, which is reflected with significant positive contributions of ADVZ to $O_3$ within the ABL in SJZ and JN. Especially in the early morning of May 20, the contribution of ADVZ to JN peaked at 38.3 ppb·h$^{-1}$ (Fig. 6). However, the negative ADVH contributions reflected the diffusion of $O_3$ by the intense horizontal winds in the cyclone system. Even at the beginning of SI, the negative effects of ADVH to $O_3$ were greater than the positive effects of ADVZ. Compared with the vertical transport of $O_3$ by the ADVZ process, the horizontal $O_3$ transport driven by ADVH occurred

earlier, which further demonstrated that the extratropical cyclone formed first and then drove the $O_3$ originating from

stratosphere to penetrate the top of ABL downward to the near-surface layer.

The simulation results provided evidence that the surface extratropical cyclone excited by the NECV delivered stratospheric $O_3$ downward to the near-surface layer increasing the near-surface $O_3$ concentrations, while the extratropical cyclone caused atmospheric horizontal motions that removed the local $O_3$. Namely, although the intrusion of stratospheric $O_3$ has the potential to augment surface $O_3$ levels, rapid dispersion and removal facilitated by the peripheral horizontal winds of

the cyclone mitigated this impact. Therefore, the strong intrusion of stratospheric $O_3$ into the near-surface layer during the SI process without surface $O_3$ concentrations exceeding the hourly standard of $O_3$ pollution of 200 $\mu g \cdot m^{-3}$ over the NCP (Fig. 5c). Furthermore, SJZ received stratospheric $O_3$ in the near-surface layer earlier than JN, and the surface stratospheric $O_3$ in these two sites peaked in the morning and evening on May 19, respectively (Figs. 6c and i). This phenomenon confirmed the results in Section 3.1 that stratospheric $O_3$ injected into the near-surface layer in the northern NCP, and then the northwest

wind of the cyclone drove the $O_3$ to drift southeastern downstream.

**3.3 Contribution of stratospheric $O_3$ from SI to NCP**

In this section, the simulations of the control experiment $CASE_{STRO3}$ and the sensitivity experiment $CASE_{noSTRO3}$ with the model WRF-Chem were applied to examine the atmospheric environmental effects of the stratospheric $O_3$ with the quantitative assessment of the contribution of stratospheric $O_3$ from the SI event. Since stratospheric $O_3$ was turned off in the

simulation of $CASE_{noSTRO3}$, the impact of the stratospheric $O_3$ during the SI event to the tropospheric $O_3$ could be assessed with the differences in chemical simulations between $CASE_{STRO3}$ and $CASE_{noSTRO3}$. Figure 7 illustrated the temporal lag changes of difference peaks in the vertical $O_3$ transport contributions to $O_3$ concentrations in the vertical altitude layers of 12–16km, 2–12km, to 0–2km to shed some light on the SI of stratospheric $O_3$ contributions from the lower stratosphere through the upper and middle troposphere into the lower troposphere including the ABL. During the SI event, the vertical $O_3$

transport contributions peaked in the lower stratosphere (12–16km) at 00:00 LST on May 19, about 9 hours late in the upper and middle troposphere (2–12km), and further 12 hour late in the lower troposphere (0–2km) influencing atmospheric environment (Fig. 7). The appearance times of these three peaks were delayed with the decreasing heights, reflecting that the SI occurrence could sequentially elevate the contributions of downward $O_3$ transport in the lower stratosphere, the free troposphere, and the ABL in the vertical cross-layer transport process of stratospheric $O_3$ during the SI event.


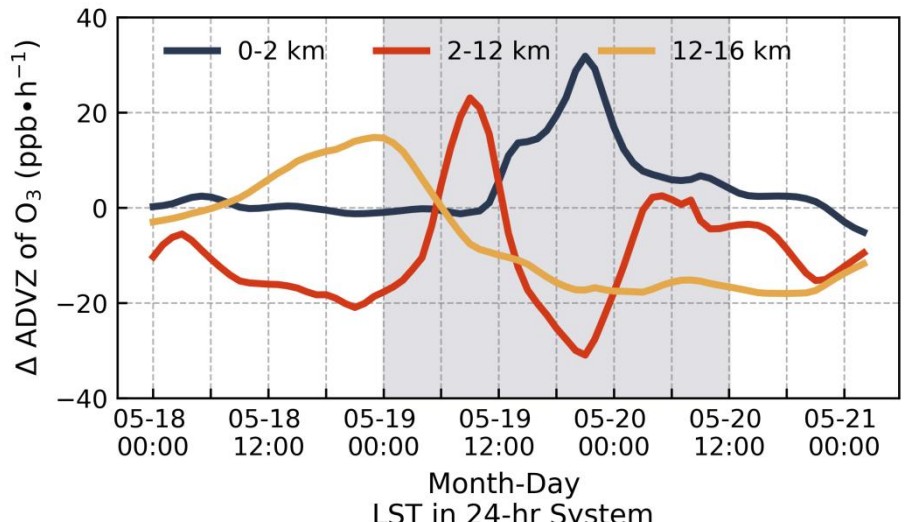

**Figure 7: The temporal variations of vertical transport rates of stratospheric O$_3$ (Δ ADVZ) in the vertical layers of 0–2 km (lower troposphere), 2–12 km (middle and upper troposphere), and 12–16 km (lower stratosphere) with the differences between CASE$_{STRO3}$ and CASE$_{noSTRO3}$ simulations with the WRF-Chem during May 18–20, 2019. The shaded area marks the SI period.**

The relative contributions of stratospheric O$_3$ to near-surface O$_3$ were calculated with dividing the absolute contributions from the simulated near-surface O$_3$ differences between CASE$_{STRO3}$ and CASE$_{noSTRO3}$ by the surface O$_3$ concentrations simulated in CASE$_{STRO3}$. Figure 8 presents the temporal variations of the stratospheric O$_3$ contributions to the near-surface layer in major urban sites in the NCP. As can be found that the intrusion of stratospheric O$_3$ sharply raised the relative contributions of O$_3$ originating from the stratosphere in the northern part of NCP, the contribution of the stratospheric O$_3$ to

near-surface O$_3$ rapidly peaked at the beginning of the SI and then gradually decreased to about 20% under the diffusion of horizontal wind (Fig. 8). It means that although the horizontal diffusion caused no remarkable increment in observed O$_3$ over the NCP during this SI event (Fig. 5), stratospheric O$_3$ contributed a relatively high percentage to the near-surface atmospheric environment. The maximum contributions of stratospheric O$_3$ to near-surface O$_3$ in these major cities ranged from 43% to 58%, with the averaged contributions from 28% to 35% in this SI event (Fig. 8). On average over the NCP

region, this SI event made the absolute contributions of 9.61 ppb to the near-surface O$_3$, accounting for 26.77% in the relative contribution. Compared with the others results, the stratospheric O$_3$ contribution in our study was lower, but the relative proportion was higher (Chang et al., 2023; Meng et al., 2022a; Meng et al., 2022b; Wang, H. et al., 2020), which could be related to the synoptic and atmospheric environmental conditions during cold vortex activity. Moreover, the peak moments of stratospheric O$_3$ contributions are delayed from north to south, reflecting the northwest wind of horizontal

circulation induced the downward stratospheric O$_3$ tongue to bend from northeast to southwest.

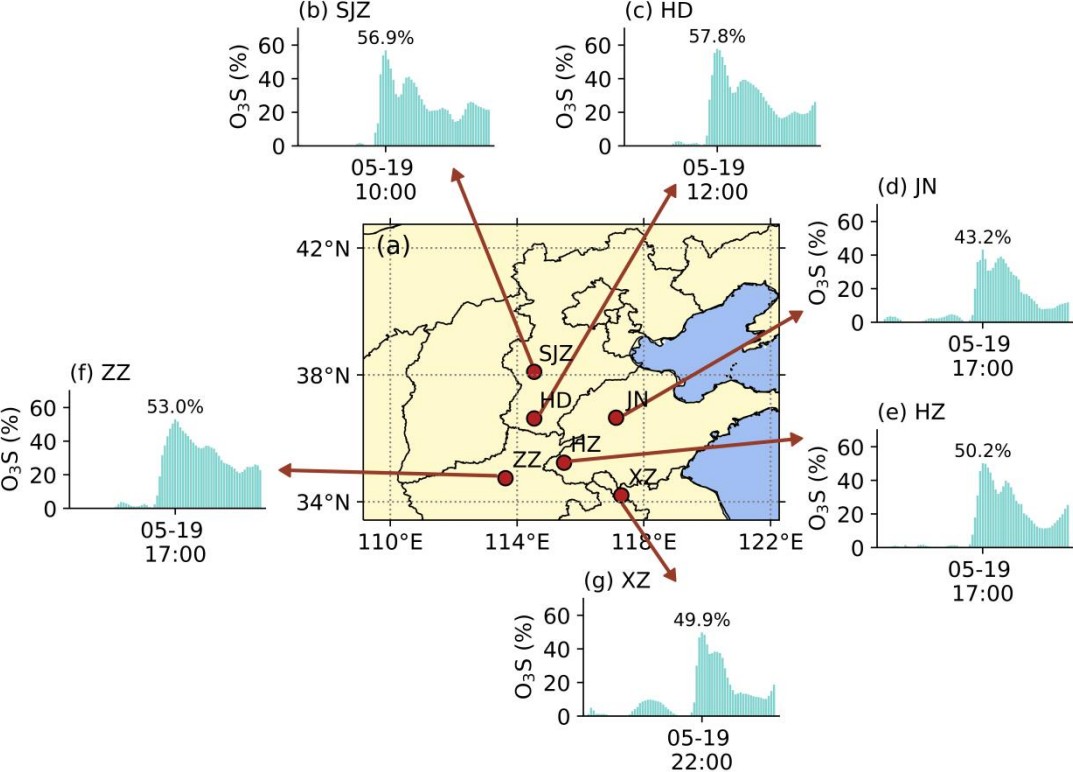

**Figure 8: Temporal variations of the simulated contribution of stratospheric O₃ to near-surface O₃ in the urban sites SJZ, HD, JN, HZ, ZZ, and XZ over the NCP. The values annotated on the bars are the maximum contributions of the stratospheric O₃ occurring at the moments (month-day / hour: minute) marked on the x-axis.**

## 4 Conclusions

The SI events could trigger the transport of O₃ from the stratosphere to the free troposphere. The SIs frequently happens around the subtropical westerlies, and the cold vortexes regulating weather and climate in mid-high latitudes are closely related to the westerly development. However, the mechanism of SI effects on the atmospheric environment with the cross-layer transport of O₃ from the stratosphere, the free troposphere to the atmospheric boundary layer has not been elucidated thoroughly. In this study, a SI process triggered by a typical cold vortex over North China was taken as an example to deeply investigate the driving mechanism of the vertical structure configuration of atmospheric circulation in the mid-high latitudes on the cross-layer transport of stratospheric O₃ to the near-surface atmosphere, and to quantify the impact of SI on the near-surface O₃ with jointly adopting the multi-source reanalysis and observation data of meteorology and environment and the numerical modeling of regional air quality.

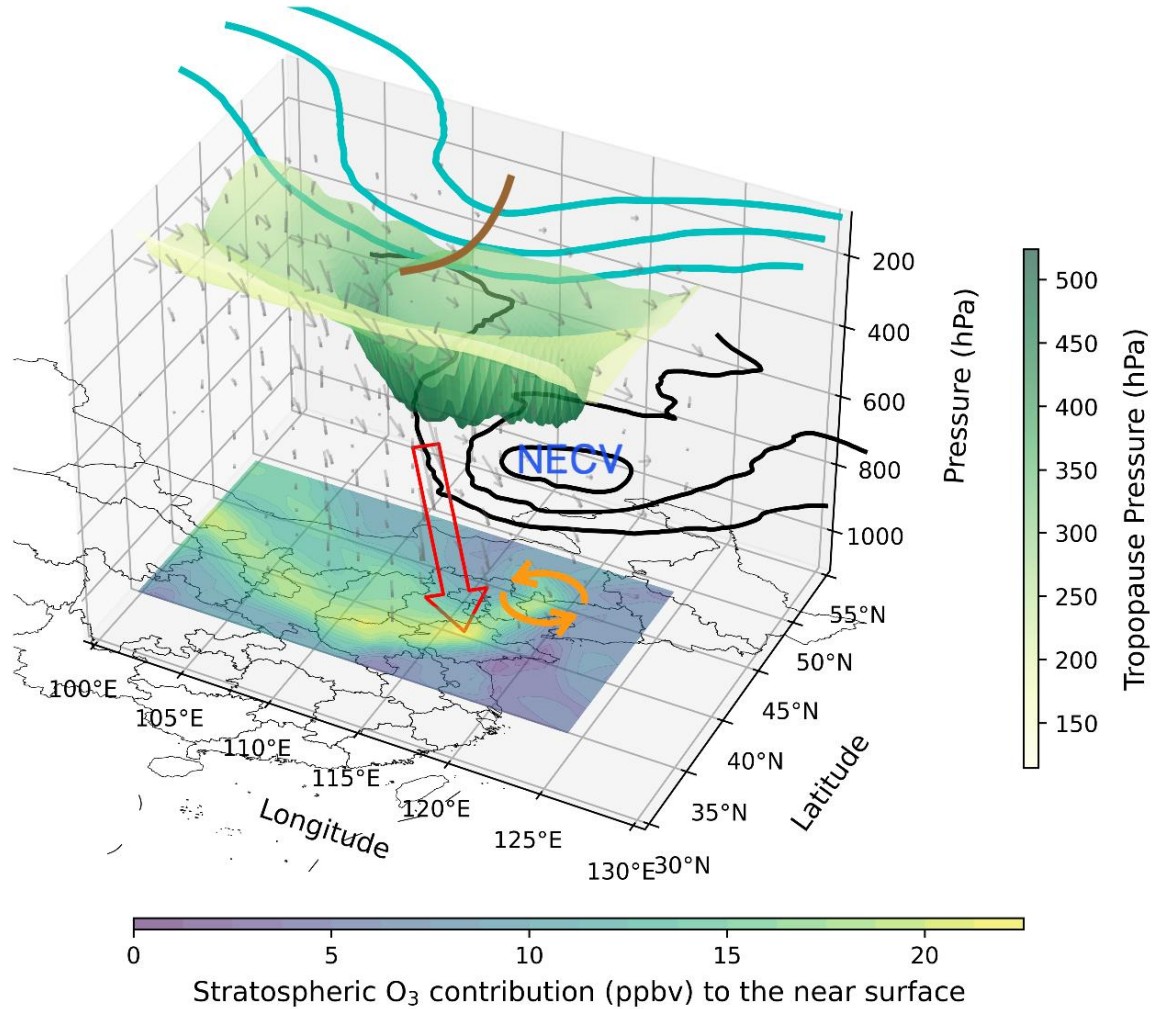

**Figure 9: Conceptual diagram of the downward O₃ transport (red arrow) of SI to near-surface layer driven by the vertical configuration of "upper westerly trough-middle NECV-lower extratropical cyclone" in the troposphere. The upper 3D surface indicates the folded tropopause in the vertical air pressures, the green and black solid lines represent the westerly trough and NECV with the geopotential height contours of 200 hPa and 500 hPa in the upper and middle troposphere, respectively. The orange cycle arrows indicate the extratropical cyclone in the lower troposphere. The shaded colors on the surface indicate the stratospheric O₃ contribution to the near-surface (ppb).**

This study revealed a mechanism of stratospheric O₃ intrusion into the tropospheric based on a case study of North China. The SI with downward transport of stratospheric O₃ to near-surface layer was driven by an extratropical cyclone system with the vertical coupling of "upper westerly trough-middle NECV-lower extratropical cyclone" in the troposphere (Fig. 9). At first, the disturbances in the westerly jet stream aroused a deep trough in the lower stratosphere and upper troposphere. The vertical shear of the jet stream on the south side of the trough caused broad-scale subsidence, then induced the tropopause folding down to 500–600hPa. The O₃-rich air from the lower stratosphere penetrated the folded tropopause into the upper and middle troposphere. As the upper trough moved eastwards and deepened southward, the westerly trough was cut off in the upper and middle troposphere, forming a typical cold vortex NECV in northern China. The compensating downdrafts of the NECV pushed the stratospheric O₃ in the upper and middle troposphere to further downward transport. Simultaneously, the NECV in the upper and middle troposphere activated an extratropical cyclone in the lower troposphere, and the downdraft on the periphery of the vertical cyclonic circulation governed the stratospheric O₃ from the free troposphere to

break through the boundary layer top invading the near-surface atmosphere.

Based on the simulation experiments, we quantified the contribution of stratospheric $O_3$ to near-surface $O_3$ during the SI process. The stratospheric $O_3$ intrusions were strengthened sequentially in the lower stratosphere, middle and upper troposphere, and lower troposphere through vertical transport, reflecting that the stratospheric $O_3$ was indeed progressively transported downwards from the lower stratosphere to the lower troposphere influencing the atmospheric environment. In this SI event, the average contribution of stratospheric $O_3$ to near-surface $O_3$ was 9.61 ppb over the NCP region, accounting for 26.77% in near-surface $O_3$ levels.

Our study illustrated the meteorological mechanism of vertical transport of stratospheric $O_3$ into the near-surface atmosphere driven by an extratropical cyclone system, improving the understanding of the influence of stratospheric $O_3$ on the atmospheric environment. The findings have significant implications for a comprehensive understanding of the cross-layer transport of air pollutants on the coordinated control of atmospheric pollution complex. The study on the mechanism of stratospheric $O_3$ intrusion into the atmospheric environment will be conducted with climatic analysis of long-term observations of meteorology and environment combined with fine simulations of atmospheric chemical and physical processes, to generalize the meteorological mechanism of stratospheric $O_3$ intrusion with climatology of stratospheric $O_3$ contribution to atmospheric environment change.

*Code and Data availability.* Code and data used in this paper can be provided upon request from Yuehan Luo (lyh_nuist@qq.com) or Tianliang Zhao (tlzhao@nuist.edu.cn).

*Author contributions.* YL, TZ, and KM conceived the analysis and modeling study. YL designed the graphics and wrote the manuscript with help from TZ, KM, JH and QY. YB, KY, WF, and CT were involved in the scientific discussion. All authors commented on the paper.

*Competing interests.* The authors declare that they have no conflict of interest.

*Financial support.* This research was supported by the National Key Research and Development Program of China (Grant No. 2022YFC3701204) and the National Natural Science Foundation of China (Grant No. 42275196).

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
