# Peer review of "A mechanism of stratospheric O3 intrusion to atmospheric environment: a case study of North China Plain"

_EGUsphere, 2024_

## Referee Comment (RC2)

Review of "A significant mechanism of stratospheric O3 intrusion to atmospheric environment: a case study of North China Plain" by Luo et al.

General comments

Stratospheric intrusion (SI) is an important process that brings ozone-rich air mass from the stratosphere to the troposphere. Among all SI events, only a limited portion can reach the surface and impact air quality. It is important, as well as challenging, to identify a SI event that reaches the surface, and to explore the associated mechanisms and impacts. Therefore, this paper is of scientific importance. The authors investigated a SI event occurring in May 2019 in North China Plain (NCP), using some of the best data and methods available. They proposed a mechanism of "upper westerly tough-middle the Northeast Cold Vortex-lower extratropical cyclone" in the troposphere to explain this SI event. The analysis showed their deep understanding of atmospheric dynamic and chemistry. The expression is logic and clear.

I am convinced of this SI event and the associated meteorological and chemistry evolution. My main concern is that surface observations (Figure 5) seem not to strongly support the claim that this SI reached the surface. The diurnal cycles of ozone and temperature seem not to be disturbed by the SI event, but the ones of carbon monoxide and humidity seem indicative. This can be due to that the average over NCP was considered. The authors are encouraged to examine the observations at individual sites for stronger indications of the SI to the surface.

Quantitatively, the authors conducted sensitivity simulations using a regional chemical transport model, WRF-Chem. They concluded that "this SI event made the absolute contributions of 9.61 ppbv to the near surface $O_3$, accounting for 26.77% in the relative contribution". However, this assessment of the SI impact seems higher than the observations showed in Figure 5 if the claim refers to the entire NCP area.

Minor comment

L22, please provide the information on latitudes, longitudes of the North China Plian, and the time the SI event occurred.

L23, remove "and".

L45, remove "the" before "stratospheric $O_3$"

L60-61, "In the mid-latitudes of the northern hemisphere, approximately 20–30% of the $O_3$ reserve in the troposphere is sourced from the stratosphere"? The number looks high.

L115, the supplement should be cited, instead of the references.

L117, leave a space between a number and its unit.

L190, at "the" before "western plateau". It is better to make the sentence clearer: western of what?

L214, "Wang, H. et al. 2020" are not found in References.

L226-227, No much change in temperature is observed in Figure 5a. Temperature seems to follow a normal diurnal cycle.

L227, change to "Precipitation".

L138-139, "Therefore, our simulation results are available and convincing." ?

L241 and L242, what kind of disturbance? Where did the disturbance come from?

L339, "significant" is used a few times in the text, also in the title and abstract. I suggest removing the word throughout. Otherwise, the authors may explain "significant" in what sense? How does this proposed mechanism compare with others? Is this a dominant or not dominant mechanism?

L339, replace "atmospheric environment" with "tropospheric".

L355, 9.61 ppbv? on average over the NCP? How large is the affected area?

Figure 4, the quality is poor, too small, not supporting the points in the text.

Figure S4, the blue lines are difficult to see.

References are not fully listed alphabetically.

It is better to indicate data sources for each figure.

Please pay attention to recent literature on the topic.

MERRA2 $O_3$ data were used substantially. However, a discussion on the data performance, especially for variable $O_3S$, is lacking.

---

## Author Comment (AC1)

Dear Editors and Reviewers,

Thank you very much for your careful review on our manuscript egusphere-2024-268. We appreciate very much your encouraging comments and constructive suggestions on improving our manuscript. We have accordingly made the careful and substantial revisions. The revised portions are marked up in the revised manuscript. Please find our point to point responses to the reviewers' comments as follows:

**Responses to the reviewer #1**

*[My main concern is that surface observations (Figure 5) seem not to strongly support the claim that this SI reached the surface. The diurnal cycles of ozone and temperature seem not to be disturbed by the SI event, but the ones of carbon monoxide and humidity seem indicative. This can be due to that the average over NCP was considered. The authors are encouraged to examine the observations at individual sites for stronger indications of the SI to the surface.]*

**Response :** Thanks to the reviewers for the valuable suggestion on our manuscript. Following the reviewers' suggestion, the changes in observed meteorological and environmental elements from the representative sites SJZ and JN (The red dots in Fig. S3) were examined in Fig. S6 with the following discussions:

The changes in observed meteorological and environmental elements from the representative sites SJZ and JN in the NCP (The red dots in Fig. S3) were examined in Fig. S6. The results showed the distinct characteristics compared with the regional averages that the diurnal cycles of $O_3$ concentrations were disturbed with the $O_3$ peaks by the SI event . The SJZ in the northwest NCP received stratospheric $O_3$ earlier and reached the spike at 10:00 LST on May 19. Then the $O_3$ concentrations gradually decreased under the influence of strong winds but still maintained a high level in the early morning of May 20. The JN in the southeast NCP was affected by the stratospheric intrusion later. While under meteorological conditions conducive to the dissipation of pollutants (wind speed up to 8 $m \cdot s^{-1}$), higher $O_3$ concentrations than the previous day were still observed, reflecting the additional contribution of stratosphere intrusion to near-surface $O_3$.

We have added the above discussion to lines 256-263 in Section 3.2 and Fig. S6 was added in the supplement.

[Figure]

**Figure S6: Hourly variations of (a, d) $T_2$, $RH_2$, (b, e) $WS_{10}$, and total cloud cover (TCC), (c, f) near-surface $O_3$ and CO concentrations in representative cities SJZ and JN from the observations in the NCP region. The shaded areas mark the periods of the SI to the near-surface layer.**

Additionally, We understand the reviewer's concern on Fig.5 that the surface observations seem not to strongly support the claim that this SI reached the surface. We explain the Fig. 5 as follows:

Once stratospheric $O_3$ transports down to the troposphere, it will undergo the identical physical and chemical processes as tropospheric $O_3$, and the conventional surface observations cannot distinguish whether $O_3$ comes from the stratosphere or is generated in the troposphere. In our proposed mechanism of "upper westerly tough-middle the Northeast Cold Vortex-lower extratropical cyclone", the vertical configuration of the entire system generated strong northwest horizontal winds (Figs. 2a-c and 4). The regional average wind speed at 10m exceeded 7 $m·s^{-1}$ and reached a maximum of 20 $m·s^{-1}$ at a single site. Although the stratospheric $O_3$ contributed 26.77 % of surface $O_3$ to the entire NCP, the fierce northwest wind quickly diffused $O_3$ (including $O_3$ originating from the stratosphere and troposphere) to downstream areas. For the near-surface $O_3$, the positive contribution of vertical stratospheric intrusion and the negative contribution of horizontal winds occurred almost simultaneously (Fig. 6), so it was difficult to observe a remarkable increment in surface $O_3$ than usual. However, under such favorable diffusion conditions, the observed near-surface $O_3$ on May 19 was slightly higher than the previous day (Fig. 5c), which hinted at the additional contribution of stratospheric $O_3$.

**Minor comments**

*[1. L22, please provide the information on latitudes, longitudes of the North China Plain, and the time the SI event occurred.*

**Response :** Many thanks for the careful comments and helpful suggestions on our manuscript. Following the reviewer's suggestion, we have added these information to lines 22 in the revised manuscript as follow:

In this study, a SI event over the North China Plain (NCP, 33–40°N, 114–121°E) during May 19–20, 2019 was taken to investigate the mechanism of the cross-layer transport of stratospheric $O_3$ with the impact on the near-surface $O_3$ based on the multi-source reanalysis, observation data and air quality modeling.

*[2. L23, remove "and".]*

**Response :** It has been corrected.

*[3. L45, remove "the" before "stratospheric O3"]*

**Response :** The word "the" before "stratospheric $O_3$" in line 48 of the revised manuscript was removed.

*[4. L60-61, "In the mid-latitudes of the northern hemisphere, approximately $20 - 30\%$ of the O3 reserve in the troposphere is sourced from the stratosphere"? The number looks high.]*

**Response :** Indeed, the principal source of surface $O_3$ is the photochemical reaction, and less stratospheric $O_3$ could reach the near-surface. While in the middle and upper troposphere, the stratosphere contributes a large amount of $O_3$, thereby increasing the stratosphere's contribution to tropospheric $O_3$ reserves.

Many previous studies have evaluated the contribution of stratospheric $O_3$ to the troposphere and obtained widely varying results. Roelofs and Lelieveld (1997) estimated that $O_3$ originating from the stratosphere contributed about 40% on average to $O_3$ in the troposphere and between 10% (in summer and at the equator)

and 60% (in winter) at the surface. These values are higher than those previously published by Austin (1991) and Follows (1992), who estimated that stratospheric $O_3$ contributed 25% at 300 hPa and less than 5% at the surface. It follows that the stratospheric contribution to tropospheric $O_3$ reserves of 20-30% is a reasonable range.

**References**

Austin, J. F., and Follows, M. J.: The ozone record at Payerne: An assessment of the cross-tropopause flux. Atmospheric Environment. Part A. General Topics, 25(9), 1873-1880, https://doi.org/10.1016/0960-1686(91)90270-H, 1991.

Follows, M. J., and Austin, J. F.: A zonal average model of the stratospheric contributions to the tropospheric ozone budget. Journal of Geophysical Research: Atmospheres, 97(D16), 18047-18060, https://doi.org/10.1029/92JD01834, 1992.

Roelofs, G. J., and Lelieveld, J. O. S.: Model study of the influence of cross-tropopause $O_3$ transports on tropospheric $O_3$ levels. Tellus B: Chemical and Physical Meteorology, 49(1), 38-55, https://doi.org/10.3402/tellusb.v49i1.15949, 1997.

*[5. L115, the supplement should be cited, instead of the references.*

**Response :** We keep the the references because the purpose of citing the literature here is to illustrate that the WRF-Chem model used in previous studies could reproduce the SI process well, rather than to evaluate the WRF-Chem simulation results in our study.

*[6. L117, leave a space between a number and its unit.]*

**Response :**we have added spaces between the numbers and units to lines 125-126 of the revised manuscript.

*[7. L190, at "the" before "western plateau". It is better to make the sentence clearer: western of what?]*

**Response :** Thanks to the reviewer for pointing out our oversight. We have modified the sentence as follow (lines 205-207 in the revised manuscript):

The mountain airflow on the lee slopes of the plateau and mountains in western NCP could also contribute to strengthening the subsidence motion in the lower troposphere (Ning et al., 2018).

*[8. L214, "Wang, H. et al. 2020" are not found in References.]*

**Response :** "Wang, H. et al. 2020" which we cited in line 314 of the original manuscript was listed on lines 509-510 of the original manuscript (lines 554-555 of the revised manuscript).

*[9. L226-227, No much change in temperature is observed in Figure 5a. Temperature seems to follow a normal diurnal cycle.*

**Response :** Yes, we agree with the reviewer's comment that no much change in temperature is observed in Figure 5a. Therefore, we proposed that "the rising air temperature required for the stronger photochemical reaction was not observed on May 19, 2019" (lines 242-243 in the original manuscript) to illustrate that the photochemical reaction might not generate more surface $O_3$ compared with the previous day.

*[10. L227, change to "Precipitation".]*

**Response:** We have corrected "the precipitation" to "Precipitation" to line 243 in the revised manuscript.

*[11. L138-139, "Therefore, our simulation results are available and convincing." ?]*

**Response :** Thanks to the reviewer for pointing out our inappropriate expression. We have corrected this sentence in lines 154-155 as follows:

All these evaluations indicate that our simulations performed well in reproducing the variations of $O_3$ and meteorological parameters during the SI process.

*[12. L241 and L242, what kind of disturbance? Where did the disturbance come from?]*

**Response :** This kind of westerly jet disturbance originates from the changes in the north-south differential in solar radiation. When air temperature difference between the north and the south gradually increases, the atmospheric baroclinicity strengthens, causing the westerly jet to shift from the zonal to the meridional circulation patterns, resulting in larger north-south airflow, eventually leading to the formation of cut-off low-pressure or blocking high-pressure systems.

*[13. L339, "significant" is used a few times in the text, also in the title and abstract. I suggest removing the word throughout. Otherwise, the authors may explain "significant" in what sense? How does this proposed mechanism compare with others? Is this a dominant or not dominant mechanism?]*
*[14. L339, replace "atmospheric environment" with "tropospheric".]*

**Response :** Thanks for the reviewer's suggestions on our manuscript. Follow the reviewer's comment, we have removed the word "significant" and replaced "atmospheric environment" with "tropospheric" in the revised manuscript.

*[15. L355, 9.61 ppbv? on average over the NCP? How large is the affected area?]*

**Response :** Yes, the amount of 9.61 ppbv is the average value for the entire NCP region during the SI process, covering from 33–40°N and 114–121°E.

*[16. Figure 4, the quality is poor, too small, not supporting the points in the text.*

**Response :** Thanks for the helpful suggestions on our manuscript. Since the selected O₃ concentration scale range in Figure 4 is large, the O₃ concentration change characteristics are not clear enough, so we narrowed the concentration scale range and redrawn Figure 4 with the better figure quality.

[Figure]

**Figure 4: Horizontal wind vectors and 24-hour changes of air temperature (shaded colors) at 950 hPa from the ERA5 data and the observed near-surface O₃ concentrations (color dots) at 16:00 LST on May (a) 18, (b) 19, and (c) 20, 2019. The red rectangles cover roughly the NCP region.**

*[17. Figure S4, the blue lines are difficult to see.]*

**Response:** Following the reviewer's comment, we have redrawn Figure S4 to make it more readable.

[Figure]

**Figure S4: Latitudinal vertical sections of O₃ concentrations (color contours) averaged over 32 °N–40 °N from the MERRA2 data during May 18–21, 2019. Black solid lines indicate the dynamical tropopause labeled by PV=2. The dashed black lines represent air temperature (℃), the solid blue lines represent relative humidity (%), and the blue rectangles mark the NCP region.**

*[18.  References are not fully listed alphabetically.*

*It is better to indicate data sources for each figure.*

*Please pay attention to recent literature on the topic.*

*MERRA2 O3 data were used substantially. However, a discussion on the data*

*performance, especially for variable O3S, is lacking.]*

**Response :** Thanks to the reviewers for pointing out our shortcomings. We have carefully checked and rearranged all cited references to ensure that they are in alphabetical order.

And we stated the data sources used in each figure and added some recent references on the topic.

Finally, we survey the works of literature and illustrate the availability of MERRA2 O₃ data. However, since O₃S is a model diagnostic variable and there are no corresponding observations, we cannot evaluate the

accuracy of the $O_3S$ data from EAC4. Follow the reviewer's comment, we added the following descriptions to lines 105-109 of the revised manuscript:

The MERRA-2 data set assimilates TCO satellite retrievals from ozone monitoring instruments and stratospheric $O_3$ vertical profiles from microwave limb sounders after 2004 (Gelaro et al., 2017; Levelt et al., 2006; Waters et al., 2006). The MERRA2 $O_3$ enables the study of SI events because it is consistent well with ozonesondes and could realistically represent the temporal and spatial variations of $O_3$, especially in the lower stratosphere and near the tropopause (Wargan et al., 2015; Wargan et al., 2017).

**References**

Chen, Z., Liu, J., Qie, X., Cheng, X., Shen, Y., Yang, M., Jiang, R., and Liu, X.: Transport of substantial stratospheric ozone to the surface by a dying typhoon and shallow convection. Atmospheric Chemistry and Physics, 22(12), 8221-8240, doi:https://doi.org/10.5194/acp-22-8221-2022, 2022.

Chen, Z., Xie, Y., Liu, J., Shen, L., Cheng, X., Han, H., Yang, M., Shen, Y., Zhao, T., and Hu, J.: Distinct seasonality in vertical variations of tropospheric ozone over coastal regions of southern China. Science of The Total Environment, 874, 162423, doi:https://doi.org/10.1016/j.scitotenv.2023.162423, 2023a.

Chen, Z., Liu, J., Cheng, X., Yang, M., and Shu, L.: Stratospheric influences on surface ozone increase during the COVID-19 lockdown over northern China. npj Climate and Atmospheric Science, 6(1), 76, doi:https://doi.org/10.1038/s41612-023-00406-2, 2023b.

Gelaro, R., McCarty, W., Suárez, M. J., Todling, R., Molod, A., Takacs, L., Randles, C. A., Darmenov, A., Bosilovich, M. G., Reichle, R., Wargan, K., Coy, L., Cullather, R., Draper, C., Akella, S., Buchard, V., Conaty, A., Silva, A. M., Gu, W., Kim, G., Koster, R., Lucchesi, R., Merkova, D., Nielsen, J. E., Partyka, G., Pawson, S., Putman, W., Rienecker, M., Schubert, S. D., Sienkiewicz, M., and Zhao, B.: The modern-era retrospective analysis for research and applications, version 2 (MERRA-2). Journal of Climate, 30(14), 5419–5454, doi:https://doi.org/10.1175/JCLI-D-16-0758.1, 2017.

Levelt, P. F., Van Den Oord, G. H., Dobber, M. R., Malkki, A., Visser, H., De Vries, J., Stammes, P., Lundell, J. O. V., and Saari, H.: The ozone monitoring instrument. IEEE Transactions on geoscience and remote sensing, 44(5), 1093–1101, doi:https://doi.org/10.1109/TGRS.2006.872333, 2006.

Waters, J. W., Froidevaux, L., Harwood, R. S., Jarnot, R. F., Pickett, H. M., Read, W. G., Siegel, P. H., Cofield, R. E., Filipiak, M. J., Flower, D. A., Holden, J. R., Lau, G. K., Livesey, N. J., Manney, G. L., Pumphrey, H. C., Santee, M. L., Wu, D. L., Cuddy, D. T., Lay, R. R., Loo, M. S., Perun, V. S., Schwartz, M. J., Stek, P. C., Thurstans, R. P., Boyles, M.

A., Chandra, K. M., Chavez, M. C., Chen, G. S., Chudasama, B. V., Dodge, R., Fuller, R. A., Girard, M. A., Jiang, J. H., Jiang, Y., Knosp, B. W., LaBelle, R. C., Lam, J. C., Lee, K. A., Miller, D., Oswald, J. E., Patel, N. C., Pukala, D. M., Quintero, O., Scaff, D. M., Snyder, W. V., Tope, M. C., Wagner, P. A., and Walch, M. J.: The earth observing system microwave limb sounder (EOS MLS) on the Aura satellite. IEEE transactions on geoscience and remote sensing, 44(5), 1075–1092, doi:https://doi.org/10.1109/TGRS.2006.873771, 2006.

Wargan, K., Labow, G., Frith, S., Pawson, S., Livesey, N., and Partyka, G.: Evaluation of the Ozone Fields in NASA's MERRA-2 Reanalysis. Journal of Climate, 2961–2988, doi:https://doi.org/10.1175/JCLI-D-16-0699.1, 2017.

Wargan, K., Pawson, S., Olsen, M. A., Witte, J. C., Douglass, A. R., Ziemke, J. R., Strahan, S. E., and Nielsen, J. E.: The global structure of upper troposphere-lower stratosphere ozone in GEOS-5: A multiyear assimilation of EOS Aura data. Journal of Geophysical Research: Atmospheres, 120(5), 2013–2036, doi:https://doi.org/10.1002/2014JD022493, 2015.

---

## Author Comment (AC2)

Dear Editors and Reviewers,

Thank you very much for your careful review on our manuscript egusphere-2024-268. We appreciate very much your encouraging comments and constructive suggestions on improving our manuscript. We have accordingly made the careful and substantial revisions. The revised portions are marked up in the revised manuscript. Please find our point to point responses to the reviewers' comments as follows:

**Responses to the reviewer #2**

*[1. Line 107: "The portion of tropospheric O3 concentrations originating from the stratosphere (O3S)", The author may need more description to substantiate this claim.]*

**Response 1:** Thanks to the reviewers for the valuable suggestion on our manuscript. According to the reviewer's comment, we added more description to substantiate the O₃S as follow (lines 111-117):

To evaluate the reproducibility of the WRF-Chem simulations for the stratospheric $O_3$ intrusion, the portion ($O_3S$) of tropospheric $O_3$ concentrations originating from the stratosphere was applied to compare with our simulation results. The stratospheric tracer tagging method in global chemistry models was used to track the transport of stratospheric $O_3$ to the troposphere by releasing stratospheric tracers (Barth et al., 2012; Chang et al., 2023). The tracer was set to 1 above the tropopause, and only physically transported and chemically decayed in the troposphere without chemical production (Chang et al., 2023; Ni et al., 2019). The $O_3S$ in the troposphere was calculated by multiplying the concentrations of $O_3$ at the tropopause and the stratospheric tracers.

**References**

Barth, M. C., Lee, J., Hodzic, A., Pfister, G., Skamarock, W. C., Worden, J., Wong, J., and Noone, D.: Thunderstorms and upper troposphere chemistry during the early stages of the 2006 North American Monsoon. Atmospheric Chemistry and Physics, 12(22), 11003–11026, doi:https://doi.org/10.5194/acp-12-11003-2012, 2012.

Chang, F., Li, J., Li, N., and Liao, H.: Stratospheric intrusion may aggravate widespread ozone pollution through both vertical and horizontal advections in eastern China during summer. Frontiers in Environmental Science, 10, 2756, doi:https://doi.org/10.3389/fenvs.2022.1115746, 2023.

Ni, Z. Z., Luo, K., Gao, X., Gao, Y., Fan, J. R., Fu, J. S., and Chen, C. H.: Exploring the stratospheric source of ozone pollution over China during the 2016 Group of Twenty summit. Atmospheric Pollution Research, 10(4), 1267–1275, doi:https://doi.org/10.1016/j.apr.2019.02.010, 2019.

*[2. Line 125 to 127: "The differences in --- in the SI event", How sensitive is the "quantitative effect" to the ozone lateral boundary conditions used in the control experiments? It is best to provide information on how to set the ozone lateral boundary conditions at the top level of the model in WRF-CHEM.]*

**Response 2:** Many thanks for the constructive suggestions on our manuscript. Since the stratospheric chemistry is not included in the WRF-Chem, an upper boundary condition (UBC) scheme derived from the Whole Atmosphere Community Climate Model was used to provide the initial and boundary chemical conditions for the stratosphere (including but not only the top level of the model) in the model (Barth et al., 2012). The UBC scheme could generate all key chemical species in the stratosphere, enabling the WRF-Chem to simulate the stratospheric intrusion processes more accurately (Barth et al., 2012; Lamarque et al., 2012; Zhao et al., 2021).

Following the reviewer's suggestion, we have added the above discussion to lines 130-134 in the revised manuscript.

More explanation about the UBC scheme here: According to the user guide of WRF-Chem, the UBC scheme will specify the $O_3$, $NO$, $NO_2$, $HNO_3$, $CH_4$, $CO$, $N_2O$, and $N_2O_5$ concentrations at the top of the model. These stratospheric concentrations override the original values as defined in the idealized chemical profile. From the top level of the model down to the tropopause, the concentrations are relaxed, using a 10-day time constant, to fixed values.

**References**

Barth, M. C., Lee, J., Hodzic, A., Pfister, G., Skamarock, W. C., Worden, J., Wong, J., and Noone, D.: Thunderstorms and upper troposphere chemistry during the early stages of the 2006 North American Monsoon. Atmospheric Chemistry and Physics, 12(22), 11003–11026, doi:https://doi.org/10.5194/acp-12-11003-2012, 2012.

Lamarque, J. F., Emmons, L. K., Hess, P. G., Kinnison, D. E., Tilmes, S., Vitt, F., Heald, C. L., Holland, E. A., Lauritzen, P. h., Neu, J., Orlando, J. J., Rasch, P. J., and Tyndall, G. K.: CAM-chem: Description and evaluation of interactive atmospheric chemistry in the Community Earth System Model. Geoscientific Model Development, 5(2), 369–411, doi:https://doi.org/10.5194/gmd-5-369-2012, 2012.

Zhao, K., Hu, C., Yuan, Z., Xu, D., Zhang, S., Luo, H., Wang T., and Jiang, R. A modeling study of the impact of stratospheric intrusion on ozone enhancement in the lower troposphere over the Hong Kong regions, China. Atmospheric Research, 247, 105158, doi:https://doi.org/10.1016/j.atmosres.2020.105158, 2021.

*[3. Line 130: "Fig.S2", Is it the average of all observations in domain 03? Please explain in detail.]*

**Response 3:** Our meteorological and environmental observation data were collected from the China Meteorological Observation Network and the National Air Quality Monitoring Network, and almost all observation stations in the networks are located in cities and towns. Therefore, we calculated the averages of meteorological elements and $O_3$ concentrations observed at all stations in the innermost domain and the averages of simulated meteorological elements and $O_3$ concentrations in the model grids corresponding to the station locations to conduct the modeling validation.

Following the reviewer's suggestions and comments, we have added the above discussion to lines 144-146 in the revised manuscript.

*[4. Line 138 to 139: "Therefore, our simulation results are available and convincing.", Please see the first question above]*

**Response 4:** Thanks to the reviewer for pointing out our inappropriate expression. We have corrected this sentence in lines 154-155 as follows:

All these evaluations indicate that our model simulations performed well in reproducing the variations of $O_3$ and meteorological parameters during the SI process.

*[5. Line 148: "Fig. S4b", Is the value of air temperature in the picture negative or positive?]*

**Response 5:** The values of air temperature in Fig. S4b are all negative, indicating the colder stratospheric air mass invade to the troposphere. We have slightly modified Figure S4 to enhance its readability:

[Figure]

**Figure S4: Latitudinal vertical sections of O₃ concentrations (color contours) averaged over 32 °N–40 °N from the MERRA2 data during May 18–21, 2019. Black solid lines indicate the dynamical tropopause labeled by PV=2. The dashed black lines represent air temperature (°C), the solid blue lines represent relative humidity (%), and the blue rectangles mark the NCP region.**

*[6. Line 148: Line 172 to 176:* "*the intense --- to the near-surface layer*", *Since both subsidence and ascending motion occur in extratropical cyclone systems, stratospheric intrusion ozone reaching the surface will rapidly diffuse and be carried back to the upper troposphere. Therefore, such events are difficult to observe.]*

**Response 6:** Thanks for the reviewer's helpful suggestions on our manuscript.

The peripheric subsidence and central ascending motion meteorologically occur in extratropical cyclones with a typical horizontal scale of 1000km. Therefore, parts of the stratospheric intrusion ozone reaching the surface can rapidly diffuse and are rarely carried back to the upper troposphere. While the center of the Northeast Cold Vortex and extratropical cyclone that prevailed the ascending motion was over Northeast China. Our studied region, the North China Plain (NCP), was located at the southwest periphery of the Northeast Cold

Vortex and the stimulated extratropical cyclone system and in the sinking zone of the vertical circulation of this system. Meanwhile, under the influence of the horizontal circulation of this system, the NCP experienced both the control of subsidence motions and the imposition of strong northwest winds. The intensity of the horizontal wind was much higher than the vertical velocity. Therefore, the invading stratospheric $O_3$ tongue tilted to the southwest and reached the surface under the comprehensive effect of vertical and horizontal winds (Figure 3), and then the stratospheric $O_3$ was gradually transported downstream in the southwest direction (Figure 8).

*[7. Line 215: "week", weak??]*

**Response 7:** Thanks to the reviewer for pointing out our oversight. We have corrected this word to line 231 in the revised manuscript.

*[8. Line 225: "Figure 5", How many weather observatories are there and how many ozone and carbon monoxide observatories are there? Why do the authors use regionally averaged observations? Have the authors looked at single-site observations of ozone in particular?]*

**Response 8:** The observed meteorological elements from 639 sites and $O_3$ and CO concentrations from 440 environmental observatories in the innermost domain (domain 03) were applied in Figure 5. Following the reviewers' suggestion, the changes in observed meteorological and environmental elements from the representative sites SJZ and JN (The red dots in Fig. S3) were examined in Fig. S6. The results showed that the diurnal cycles of $O_3$ concentration presented noteworthy characteristics compared with the regional averages. The SJZ in the northwest received stratospheric $O_3$ earlier and reached the spike at 10:00 LST on May 19. Then the $O_3$ concentrations gradually decreased under the influence of strong winds but still maintained a high level in the early morning of May 20. The JN city in the southeast was affected by the stratospheric intrusion later. While under meteorological conditions conducive to the dissipation of pollutants (wind speed up to 8 $m \cdot s^{-1}$), higher $O_3$ concentrations than the previous day were still observed, reflecting the additional contribution of stratosphere intrusion to near-surface $O_3$.

We have added the above discussion to lines 256-263 in Section 3.2 and Fig. S6 was added in the supplement.

Furthermore, we added the number of meteorological and environmental observation sites to line 109 in the revised manuscript.

[Figure]

**Figure S6: Temporal variations of (a, d) $T_2$, $RH_2$, (b, e) $WS_{10}$, and total cloud cover (TCC), (c, f) near-surface $O_3$ and CO concentrations in representative cities SJZ and JN from the observations in the NCP region. The shaded areas mark the periods of the SI to the near-surface layer.**

*[9. Line 258:"vertical mixing (VMIX) --- vertical advection (ADVZ)", What is the difference between VMIX and ADVZ?]*

**Response 9:** In the integrated process analysis of the WRF-Chem model, the VMIX term represents the impact of vertical entrainment mixing caused by turbulent motion on pollutants, which mainly occurs within the atmospheric boundary layer. The ADVZ term reflects the vertical advection transport of pollutants driven by vertical wind.

*[10. Line 266:"Fig. 6", How did the authors choose the IPR times shown in Figure 6 for SJZ and JN?]*

**Response 10:** The horizontal northwest wind drives the stratospheric $O_3$ that invades the surface to present notable characteristics of downstream transport (Figure S3), causing the temporal variation of the contributions of ADVZ to $O_3$ in the boundary layer to be unsynchronized in SJZ and JN. Therefore, we chose different times to discuss the IPR in SJZ and JN. Furthermore, we considered that discussing the IPR at all times during the stratospheric intrusion would be a little redundant in plotting and writing. Therefore, based on the temporal variations of the simulated contribution of stratospheric $O_3$ to near-surface $O_3$ in SJZ and JN (Figure 8b and d), we selected several discontinuous but representative times that can reflect the temporal variation characteristics of the IPR during the intrusion process to conduct the discussions.

*[11. Line 280: "without o3 pollution", confused]*

**Response 11:** Thanks for the helpful comment on our manuscript. We have deleted "without $O_3$ pollution" in the revised manuscript. What we want to express here is:

Although the intrusion of stratospheric $O_3$ has the potential to augment surface $O_3$ levels, rapid dispersion and removal facilitated by the peripheral horizontal winds of the cyclone mitigated this impact. Therefore, steep rises in surface $O_3$ concentrations were conspicuously absent during the SI process, and no sustained regional $O_3$ pollution emerged over the NCP. This conclusion could be reflected in Figure 5c, which shows that the regional averaged $O_3$ concentration over the NCP was around 100 $\mu g \cdot m^{-3}$ during the SI period. Only the hourly $O_3$ concentrations in part of the sites exceeded 160 $\mu g \cdot m^{-3}$ temporarily, and none of them reached 200 $\mu g \cdot m^{-3}$ exceeding the standard of $O_3$ pollution.

According to the reviewer's comment, we have added the following discussion to lines 302-305 of the revised manuscript:

Namely, although the intrusion of stratospheric $O_3$ has the potential to augment surface $O_3$ levels, rapid dispersion and removal facilitated by the peripheral horizontal winds of the cyclone mitigated this impact. Therefore, the strong intrusion of stratospheric $O_3$ into the near-surface layer during the SI process without surface $O_3$ concentrations exceeding the hourly standard of $O_3$ pollution of 200 $\mu g \cdot m^{-3}$ over the NCP (Fig. 5c).

*[12. Line 290: "Figure 7", Is it the model simulation result on domain 03?]*

**Response 12:** Yes, Figure 7 presents the simulated differences in ADVZ contribution to $O_3$ in different vertical layers between the innermost domain of CASE$_{STRO3}$ (the control experiment) and CASE$_{noSTRO3}$ (the simulation experiment), indicating the temporal variations of stratospheric $O_3$ transport to various atmospheric layers in the troposphere.

---

## Author Response (AR2)

Dear Editors and Reviewers,

Thank you very much for your careful review on our manuscript egusphere-2024-268. We appreciate very much your encouraging comments and constructive suggestions on improving our manuscript. We have accordingly made the careful and substantial revisions. The revised portions are marked up in the revised manuscript. Please find our point to point responses to the reviewers' comments as follows:

**Responses to the reviewer #1**

*[ "Quantitatively, the authors conducted sensitivity simulations using a regional chemical transport model, WRF-Chem. They concluded that "this SI event made the absolute contributions of 9.61 ppbv to the near surface O3, accounting for 26.77% in the relative contribution". However, this assessment of the SI impact seems higher than the observations showed in Figure 5 if the claim refers to the entire NCP area."*

*Can the authors also address my 2nd concern? I understand that the relative contribution of "26.77%" value is a simulated result. The authors can have some discussion about this estimation.]*

**Response :** Thanks to the reviewer for the comments on our manuscript.

We understand the reviewer's concern on Fig.5 that the contribution of 26.77% of the SI impact seems higher than the observations showed in Fig. 5. We have explained the Fig. 5 in the revised manuscript (lines 256-261 and lines 339-343) as follows:

Although stratospheric $O_3$ intensely invaded the near-surface over the NCP region, the near-surface $O_3$ was strongly diffused downstream by the northwest wind. Meanwhile, the horizontal diffusion also prevented $O_3$ from the stratosphere from accumulating over the NCP (Figs. 2a-c and Fig. 4). Also, the routine ground observations cannot distinguish whether $O_3$ comes from the stratosphere or local generation. However, the near-surface $O_3$ observed on May 19 was slightly higher than the previous day under such facilitated diffusion conditions (Fig. 5c), which proves that the SI exerted additional contributions on the near-surface $O_3$ over the NCP region.

In addition, in terms of the simulation results, the contribution of the stratospheric $O_3$ to near-surface $O_3$ rapidly peaked at the beginning of the SI and then gradually decreased to about 20% under the diffusion of

horizontal wind (Fig. 8). It means that although the horizontal diffusion caused no remarkable increment in observed $O_3$ over the NCP during this SI event, stratospheric $O_3$ contributed a relatively high percentage to the near-surface atmospheric environment.